# The Switch to Online Learning during the COVID-19 Pandemic: The Interplay between Personality and Mental Health on University Students

**DOI:** 10.3390/ijerph20075255

**Published:** 2023-03-24

**Authors:** Marianna Alesi, Giulia Giordano, Ambra Gentile, Barbara Caci

**Affiliations:** Department of Psychology, Educational Sciences and Human Movement, University of Palermo, 90128 Palermo, Italy

**Keywords:** mental health, online learning, motivation, personality, self-efficacy, COVID-19, anxiety, neuroticism, cluster analysis

## Abstract

The switching from traditional to online learning during the COVID-19 pandemic was challenging for students, determining an increase in physical and mental health problems. The current paper applied a two-step cluster analysis in a large sample of *n* = 1028 university students (Mage = 21.10 years, SD = 2.45 years; range: 18–30 years; 78.4% females). Participants responded to an online survey exploring neuroticism, trait/state anxiety, general self-efficacy, academic motivation, fear of COVID-19, the impact of the COVID-19 pandemic on physical and mental health, and the help requests. Results showed two significant clusters of students having a Maladaptive Academic Profile (*n* = 456; 44.4%) or an Adaptive Academic Profile (*n* = 572; 55.6%). Significant differences were found between the two clusters, where students belonging to the Maladaptive Academic Profile reported higher levels of neuroticism, higher dispositional and situational anxiety, and fear of COVID-19, and lower self-efficacy and academic motivation than students of the Adaptive Academic Profile cluster. In addition, more physical or mental health problems and help requests, mainly to partners during the COVID-19 pandemic, were found in the Maladaptive Academic Profile cluster compared to the Adaptive Academic Profile. Finally, the practical implications of the study’s results in implementing university counseling services as protective measures to contrast psychological distress in the long-term COVID-19 pandemic are discussed.

## 1. Introduction

The university developmental age, corresponding to young adulthood (18–25 years old), is characterized by developmental tasks aimed at experiencing and managing independence and autonomy and appears to be a developmental phase susceptible and at risk of developing psychological problems such as anxiety and depression [1,2].

The scientific literature has largely demonstrated how the COVID-19 outbreak, regardless of country, negatively influenced college students’ mental health with an escalation of anxiety, depression, disruption in sleep, fears, loneliness, and warning up to extreme conditions such as post-traumatic stress disorder (PTSD) [3,4,5]. For example, 90.879 college students from different countries (China, India, Israel, Jordan, Saudi Arabia, Turkey, France, Greece, Italy, Russia, Belarus, Albania, South and North America, Ethiopia, and Malaysia) showed an increase in anxiety (39.4%), depression (31.2%), and stress (26.0%) compared to the pre-pandemic period (22.1% anxiety, 19.7% depression, and 13.4% stress). The highest prevalence of anxiety levels (34.6% vs. 22.9%) and depression (32.4% vs. 26.0%) was retrieved in females compared to males [6]. During the COVID-19 pandemic, anxiety had the highest prevalence among mental health disorders in university students [7,8,9,10,11]. According to [8], the impact of the COVID-19 pandemic on university education and performance can be explained by referring to the close relationship between educational outcomes and the mental well-being of university students. In youths’ lives, the university has a predominant role in social acceptance and relationships, educational aspirations, academic engagement and success, and expectations for future employment [12]. The above-mentioned factors created additional distress that added to students’ fears and mental health problems, adversely affecting their learning. The extreme consequence was spilling heavy stress into academic burnout [13]. Three components describe this psychological syndrome: emotional exhaustion, cynicism, and a low sense of accomplishment [14]. Academic burnout is caused by long-term and extreme academic pressure. It develops when educational resources are insufficient to cope with the demands/risk factors that considerably enhance stress levels with consequences resulting in the decline of enthusiasm and enjoyment for learning and, sometimes, academic failure. However, findings on changes in anxiety levels are mixed and support an increase in the severity of anxiety [15,16] or a decrease after two weeks of confinement measures [17]. Longitudinal research demonstrated a reduction of anxiety symptoms from 20.2% of student-reported moderate-to-severe anxiety symptoms during the first COVID-19 wave to 15.6% during the second wave in 676 university students, decreasing infections and deaths [18]. The different levels of anxiety can be explained by sociodemographic risk factors, such as gender, age, and living away from their family or country, and by psychological variables, such as a neurotic personality profile [19], level of trait anxiety [20], a maladaptive motivational profile [21], or different levels of self-efficacy [22]. Concerning sociodemographic factors, female students showed more anxiety than males, probably for a worse perception of self-efficacy and self-esteem [23,24]. On the other hand, older students were more anxious because they experienced more pressing concerns related to the risk of delay in the conclusion of their academic career and starting a professional career [3,23]. Furthermore, the pandemic stress caused higher rates of state anxiety in students with higher baseline trait anxiety or pre-existing diagnosis of psychiatric disorders. Moreover, students’ stress was negatively correlated to academic self-efficacy [20]. As demonstrated by research, a close relationship between self-efficacy, academic performance, and educational aspirations influences the university course of study. These relations are mediated by motivational factors such as task choice, effort, persistence, resilience, and academic expectations [25]. In a stressful situation such as the COVID-19 pandemic, the profile of flourishing university students (22.3% of 1124 Italian university students) was characterized by low efficacy beliefs, low academic satisfaction, and awareness of the risk of COVID-19 [12]. Negative emotions raised by stress and tension about students’ pandemic experiences and concerns could have reduced achievement and self-efficacy [21]. University students with higher anxiety levels appeared less motivated, discouraged, and bored, adversely affecting self-efficacy and subsequent academic aspirations [22,26]. Moreover, self-efficacy showed a mediating effect on university students in the association between COVID-19 contagion concerns and depression. Students who were self-perceived as less able to manage negative emotions were less likely to border concerns about COVID-19 contagion but were more likely to refer to depressive symptoms [27]. Another variable suitable to influence the perception of the situation as more threatening is personality [28]. Individuals, having personality profiles characterized by neuroticism, worrying, emotional variety, and insecurity, were more likely to experience higher levels of distress, negative emotions, and cognitions [29]. Following the Five Factors Model [30], neuroticism is the most relevant personality trait for facing fears because it enhances the perception of risk, emotional reactivity (i.e., anger, sadness, anxiety), the sensitivity to punishment signals, and decreases the resources to manage and cope with stressful events. Specifically, higher levels of neuroticism are related to significant fear about the impact of COVID-19 on daily life, a predisposition to generalize and experience negative emotions, more boredom, and adverse fantasies [19]. People, predominantly female, with higher neuroticism displayed more worries about the ongoing COVID-19 pandemic than those with lower neuroticism [4]. Extensive research has demonstrated that anxiety is indirectly related to academic success. The feeling of fear or apprehension when students face stress conditions can negatively affect their intellectual capacity to focus on homework, concentrate, learn, memorize, solve questions, have quick reaction times, complete schoolwork, receive good marks, and prepare for final exams [20,31,32,33]. Nevertheless, university students showed concerns about their academic performance and decreased motivation. The uncertainty about the future and the evolution of the COVID-19 pandemic was a potent threat to competence and autonomy needs. The lack of relationships with peers and students threatened the need for relatedness supported by cooperation tasks and group dynamics [20,33,34]. Finally, the fear of COVID-19 is another variable correlated with anxiety, depression, and stress symptoms in university students [35,36].

However, the literature mentioned above is heterogeneous across or within a series of socio-demographic, psychological, or clinical variables. In an informative way, it is still a matter of debate. Therefore, using well-suited statistical techniques to achieve stable and robust conclusions on this issue also appears critical. Clustering techniques can serve this purpose by identifying homogeneous subgroups presenting similar characteristics within a large sample [37]. Amongst the several approaches available, the two-step cluster analysis [38,39] appears to be well suited for psychological data, as it handles ordinal as well as nominal variables, which can be more informative for educational or clinical practice [40]. Indeed, data obtained from classical self-report questionnaires are not purely quantitative and are better represented as ordinal measures, i.e., classifying subjective performance according to ordinal values that specify whether the score is “above,” “within,” or “below” the normative range. Nevertheless, hierarchical or k-means methods present several limitations, such as applicability to continuous variables only, the assumption of normality of distribution, and an arbitrary choice of the number of clusters [41,42]. Based on these considerations, the main goal of the present study was to depict the psychological profiles of university students forced to change their academic learning routines from face-to-face to OL due to the COVID-19 pandemic. It must be noted that the current study was performed in the second Italian lockdown phase (March–May 2021) which persisted in restricted measures of social distancing and isolation due to the COVID-19 pandemic. This choice allows us to evaluate the effects of more restrictive lockdown phase 1 on the studied dimensions, after one year, and identify how psychological variables such as personality, trait and state anxiety, self-efficacy, academic motivation, and fear of COVID-19 were related to the negative consequences of the COVID-19 pandemic on physical and mental health as to behavioral patterns to ask someone for help. The study is cross-sectional in nature and aims to describe different profiles emerging on the studied variables in a sample of university students, applying cluster analysis to identify homogeneous subgroups presenting similar characteristics within a large sample. No similar study (to our knowledge) was performed yet in Italy.

## 2. Materials and Methods

### 2.1. Participants

One thousand twenty-eight university students aged 18–30 years (M_age_ = 21.10 years, SD = 2.45; 78.4% females) were recruited from the University of Palermo using a snowballing procedure to participate in a survey. The survey link was posted on the virtual classrooms of the first and second-degree courses researchers’ OL university courses and some social media pages of students’ associations over twelve weeks during the COVID-19 second-wave Italian lockdown phase (March–May 2021). The inclusion criterion for the survey was actively attending an online university course. According to the Declaration of Helsinki, participants gave written consent about the anonymity of data handling. All participants were not compensated either financially or through additional university credits. They completed the assessment procedure of the study in their online classrooms with an average time of about 30 min. Data were collected automatically by MS Forms when participants filled out an electronic version of a socio-demographic questionnaire and self-report measures of personality, state-trait anxiety, self-efficacy, academic motivation, fear of COVID-19, and the effect of COVID-19 on physical and mental health. The Bioethics Committee of the University of Palermo has approved the current study (n. 38/2021).

### 2.2. Measures

#### 2.2.1. Socio-Demographic Questionnaire

Participants filled out a socio-demographic questionnaire comprising three sections collecting data about, respectively, (1) student’s profile (e.g., gender, age, course type, degree, areas of study, course year, status, living in a big/small city); (2) study habits and home/OL study environment (e.g., studying at home or in other places, studying alone or with roommates/colleagues, interaction with teachers during OL lessons) during the COVID-19 pandemic; and (3) academic achievement measured as the average score of the entire curriculum.

#### 2.2.2. Personality Inventory (PI)

The Personality Inventory (PI) [43] is a questionnaire consisting of 20 items and 5 subscales, with each of the 4 items related to the five personality factors. Based on the FFM [30], the PI measures Neuroticism (N) as linked to emotional instability, Conscientiousness (C) to efficiency and diligence at work, Extraversion (E) to trendiness to sociability, Agreeableness (A) to cooperation with others and trusting, and Openness (O) to culture and experience. Each item is rated on a 5-point scale (from 1: Strongly disagree to 5: Strongly agree). In the present study, we computed the total score only for the N scale by averaging scores for each item of the scale items (Cronbach’s alpha coefficient was 0.75).

#### 2.2.3. State-Trait Anxiety Inventory

In this study, we adopted an Italian translation of the short-term version of the State-trait Anxiety Inventory which contains 2 independent subscales consisting of 10 items [44]. Five items measure state anxiety (STAI-S) (e.g., I feel that difficulties are piling up so that I cannot overcome them), and the other five items assess trait anxiety (STAI-T) (e.g., I feel confused). Each item is rated on a 4-point scale (1: Not at all, 2: Somewhat, 2: Moderately so, 4: Very much so). Both subscales are assessed separately. A high score indicates high anxiety. Cronbach’s alpha coefficients of STAI were 0.84 for STAI-T and 0.88 for STAI-S.

#### 2.2.4. The General Self-Efficacy Scale

This study used the Italian Adaptation of the General Self-Efficacy Scale [45], comprising items with good internal reliability (Cronbach’s alphas between 0.76 and 0.90) and validity. Each item is rated on a 4-point scale (from 1: Not at all true to 4: Exactly true). The total score was calculated by finding the sum of all items for each scale. It ranges between 10 and 40; a high score defines a high general self-efficacy. In line with the literature (Scholtz et al., 2002), the standardized Cronbach’s α coefficient of self-efficacy in the present study was 0.84.

#### 2.2.5. Academic Motivation Scale (AMS-C 28)

The Academic Motivation Scale (AMS-C 28) [46] measures extrinsic, intrinsic motivation, and amotivation. As stated by self-determination theory, motivation was measured across several academic disciplines [47,48]. The current study was administered the Italian version of the AMS, provided by [49]. It was composed of five subscales: Amotivation (e.g., I once had good reasons for going to college: however, now I wonder whether I should continue do not know), External Regulation (e.g., Because with only a high-school degree, I would not find a high-paying job later on), Introjected Regulation (e.g., To prove to myself that I am capable of completing my college degree), Identified Regulation (e.g., Because eventually, it will enable me to enter the job market in a field that I like), and Intrinsic Regulation (e.g., Because I experience pleasure and satisfaction while learning new things). Each subscale consists of four items rated on a 4-point scale (from 1: Not at all true to 4: Exactly true). Following the AMS scoring method stated in Howard and colleagues’ meta-analysis [50], we computed the relative autonomy index (RAI) to measure the person’s overall motivational orientation. A more autonomous regulation is represented by positive scores on the RAI index, while negative scores define a more controlled regulation. Therefore, we calculated the following formula: RAI = (+2 × Intrinsic Motivation subscale score) + (+1 × Identified Regulation subscale score) + (−1 × External Regulation subscale score) + (−2 × Amotivation subscale) to assign a different weight to each AMS subscale score. According to [51], the Introjected Motivation subscale was not considered in computing RAI (Cronbach’s alpha 0.86).

#### 2.2.6. Fear of COVID-19

The Fear of COVID-19 (FCV-19S) [52] is a self-report scale composed of 7 items. Each item is scored on a 5-point scale (from 1: strongly disagree to 5: strongly agree). This scale measures the fear of COVID-19 in adults (e.g., I am most afraid of COVID-19). The total score was calculated by averaging each participant’s score for each item (Cronbach’s alpha = 0.85).

#### 2.2.7. Impact of COVID-19 on Physical and Mental Health and Help Requests

The impact of COVID-19 on physical and mental health was measured by asking participants a direct question (i.e., Do you have Physical and Mental Health problems during the COVID-19 pandemic?) and then specifying the physical and mental health problems, such as organic and functional problems (e.g., gastro/intestinal problems, respiratory problems, headaches), anxiety, insomnia, mood alteration, inappetence, and asthenia. The help requests were assessed by inviting participants to indicate first if they asked for help or not (i.e., Do you ask someone for help during the COVID-19 pandemic?) and then to specify whom (i.e., Whom you ask for help?) choosing from different categories of people such as partner, relatives, friends, psychologists, doctors, spiritual guide, or priests.

### 2.3. Statistical Analysis

All statistical analyses were computed using SPSS software version 26.0 (IBM Corp. Released 2019). Descriptive statistics (frequencies, means, standard deviations, skewness, and kurtosis), multivariate analysis of variance (MANOVA), and Pearson’s bivariate correlation for all the study variables were calculated in the preliminary analyses. Then, a cluster analysis was used to identify the psychological profiles of university students during the OL activities due to the COVID-19 pandemic. Cluster analysis is recommended for segmenting populations because it addresses the multidimensional nature of the construct. Thus, this method allows dividing the respondents into groups exhibiting maximum within-group similarity and between-group differences based on the assumed criteria [53]. Such a technique presents several advantages compared to traditional data reduction methods, such as determining the number of clusters based on a statistical measure of fit (AIC or BIC) rather than an arbitrary choice. It also allows for using categorical and continuous variables simultaneously, analyzing atypical values (i.e., outliers), and being able to handle large datasets [38,39,40,41]. The present study applies the two-step cluster analysis, a hybrid statistical approach that first uses a distance measure to separate groups and then a probabilistic approach (such as latent class analysis) to choose the optimal subgroup model [40]. The first step was to identify pre-clusters. The second step was to refine this initial estimate by finding the most significant increase in distance between the two closest clusters in each hierarchical clustering stage. This study used log-likelihood distances for continuous variables to specify fixed clusters. All criteria variables were transformed into z-scores. To assess the optimal number of clusters in our analysis the Auto-Clustering statistic in SPSS output was used. An average silhouette measure of cohesion and separation was applied to assess the obtained solution’s quality. This measure reflects the efficacy of a cluster solution in maximizing within-cluster homogeneity and between-cluster heterogeneity. An average silhouette coefficient of 0.5 indicated a reasonable solution; less than 0.2 indicated a problematic solution [54]. Furthermore, the ratio of the sizes between the most significant cluster and the smallest cluster was calculated. The expected value of the ratio of sizes was below 2, but it should not exceed 3 [53].

A MANOVA was used to examine cluster differences. The effect size was calculated as a partial eta squared, following Cohen’s reference points to define small (η^2^ = 0.01), medium (η^2^ = 0.06), and large (η^2^ = 0.14) effects [55].

## 3. Results

### 3.1. Preliminary Analysis

Descriptive statistics show that our sample is mainly composed of female (78.4%) in-course students (96.2%) attending the first year of bachelor’s degree courses (60.5%) in the humanities area (59.6%) with a mean age of 21.10 years (SD = 2.45). During the COVID-19 social restriction measures, a high percentage of students lived in big city apartments (48.8%), but a minority were in small city areas (26%), even if all were with their parents (94.1%). As regards academic achievement, almost all students affirmed they had performed online examinations during the OL courses (98.2%), but only 64.3% of participants could follow the exam schedule. On average, they obtained a score of 27.07/30.0, which equals a B level in the US system. Additionally, 61.3% of students request help from partners (19.7%), friends (14.9%), relatives (13.1%), health professionals likewise, psychologists or medical doctors (9.7%), and spiritual directors (0.4%).

Table 1 shows the means, standard deviations, skewness, kurtosis, and Pearson’s bivariate correlations for all the psychological variables. Descriptive statistics show that some observed variables were not normally distributed with skewness and kurtosis values >1.00 [56]. However, Pearson’s bivariate correlations show significant associations among all the studied variables.

Results about the negative consequences of the COVID-19 pandemic on physical and mental health show that participants suffered mainly from anxiety (46.1%), mood alteration (43.5%), and insomnia (35%), even if 36% of participants affirmed to have any problems.

### 3.2. Cluster Analysis

A two-step cluster analysis was conducted to identify the psychological profiles of university students during the OL activities due to the COVID-19 pandemic. Specifically, we considered the participants’ scores at all the psychological measures and their answers about the impact of COVID-19 on physical and mental health and the help requests (nine variables in total).

The Auto-Clustering statistics table in SPSS output was used to assess the optimal number of clusters in our analysis, as shown in Table 2.

Although the lowest BIC coefficient is for five clusters, according to the SPSS algorithm, the optimal number of clusters is two, because the largest ratio of distances is for two clusters.

Based on the analysis, the two clusters were identified: Cluster 1 (*n* = 456; 44.4%) and Cluster 2 (*n* = 572; 55.6%). In addition, the analysis showed 0.3 points for the average silhouette measure of cohesion and separation, indicating that the solution is acceptable. Moreover, the ratio of the sizes between the most significant and smallest clusters was calculated, finding a value of 1.25, which is good, too [53].

As shown in Figure 1, the two clusters of students present very different profiles based on their z-scores. Indeed, the results depicted a sort of specular pattern in which Cluster 1, named Maladaptive Academic Profile, comprises high neurotic students high both on dispositional and situational anxiety and on the Fear of COVID-19 scale. They have low levels of self-efficacy and are more extrinsically motivated in pursuing their academic goals. They affirmed physical or mental health problems and requested mainly partners to help during the COVID-19 pandemic.

Conversely, Cluster 2, named Adaptive Academic Profile, has very emotionally stable students with low levels of dispositional and situational anxiety and fear of COVID-19. However, they have higher levels of self-efficacy and are more intrinsically motivated to pursue their academic goals. They affirmed not having physical or mental health problems and not requesting anyone for help during the COVID-19 pandemic.

For both clusters, the most important predictor was to have physical and mental health problems (above 0.8).

Then, a MANOVA was performed considering the two clusters as between-subject factor variables and scores at psychological measures as dependent variables.

The results showed a significant effect for the between-subject factor CLUSTER, F (6, 1021) = 49.36, *p* < 0.001. Additionally, significant between-subjects effects have emerged for all the dependent variables as summarized in Table 3.

As shown in Figure 2, large effect sizes were found for neuroticism and state and trait anxiety; medium effect sizes were found for self-efficacy and fear of the COVID-19 pandemic; and small effect sizes were found for academic motivation.

## 4. Discussion

The main goal of the present study was to depict university students’ psychological profiles during the period of forced change in their academic learning routines from traditional face-to-face classes to online learning due to the Italian lockdown phase 2 of the COVID-19 pandemic. To this aim, the study adopted a descriptive approach based on bivariate correlations and a two-step cluster analysis. Our goal was to apply a segmentation on a sample of university students based on psychological variables such as personality, trait and state anxiety, self-efficacy, academic motivation, and fear of COVID-19 and assess different clusters related to the negative consequences of the COVID-19 pandemic on physical and mental health as to behavioral patterns to ask someone for help. Data were collected in the period between March and May 2021 when in Italy restricted measures of social distancing and isolation due to the COVID-19 pandemic persisted.

Results of Pearson’s bivariate correlations showed a strict association among all the study variables in line with the well-known psychological literature. They gave information about the consistency of the measures adopted in the current study. Moreover, the results of the two-step cluster analysis identified two homogeneous subgroups presenting similar characteristics, defining students with a Maladaptive or an Adaptive Academic Profile.

For the whole sample, Pearson’s bivariate correlations evidenced that the dispositional trait of neuroticism is positively related to state and trait anxiety, as well as to fear of COVID-19, and this outcome is in line with the Five Factors Model [30]. Additionally, the results of the two-step cluster analysis evidenced that the two clusters differed in scores at dispositional traits of neuroticism, trait and state anxiety, and fear of COVID-19 measures. These results are coherent with the psychological literature about the strict interdependence of neurotic traits and situational anxiety [19,57], showing that the dispositional trait of neuroticism leads people to experience more than stressful and unpleasant emotions, such as anger, anxiety, depression, or vulnerability [30]. Previous studies have also found that neuroticism is the most significant trait that predicts more robust conditioned fear responses in the general population [58]. The descriptive definition of neurotic personality evidenced that highly neurotic people have scarce resources to manage stress [59], thus during the uncertain situation, likewise the COVID-19 pandemic, where people might perceive to have any control over the risk of Sars-Cov2 infection, they are more likely to develop fear [60]. The psychological literature on personality shows that neuroticism and fear of unpredictable events increase the degree of generalized fear [61]. Similarly, neuroticism is related to adverse emotional outcomes in stressful life experiences [62], so high neurotic individuals also have a high susceptibility to psychological distress, and are more prone to experiencing anxiety, anger, and sadness [30]. The data about the fact that anxiety affects individuals’ health and well-being during times of infectious epidemic crisis are also reported both in pre-pandemic studies [63] and in studies performed during the COVID-19 pandemic [19,64,65]. Many studies reported that the negative consequences of the coronavirus pandemic on people’s mental health refer to elevated symptomatology levels in anxiety, general stress, depression, post-traumatic stress [66], unpleasant emotions, anger, vulnerability [67], loss of vitality [68], lack of energy, an inability to start and carry out daily activities, and difficulty concentrating at schools or work [69]. Consistently, our findings show that students of the Maladaptive Academic profile reported more physical or mental health problems such as anxiety, mood alteration, and insomnia.

Similarly, the literature on online learning reported different effects for the introduction of online learning in the university learning context. Indeed, OL was challenging for many university students because of the lack of online experiences, resources such as Internet connectivity or digital devices, and a suitable home study environment [20]. However, other studies evidenced indirect consequences for OL showing that university students experienced an increase in problematic use of social media by enhancing time usage and frequency [70], a decrease in their academic achievement [71], and in many cases, mental health [33]. Even if the current study did not aim to identify dispositional, situational, or learning contextual predictors impacting the mental health of university students, the cluster analysis depicted an interplay of multiple individual, social, and learning factors concurring together in the process.

Indeed, significant negative associations arose between neuroticism, state/trait anxiety, fear of COVID-19, self-efficacy, and academic motivation. Such results at bivariate associations were corroborated by cluster analysis, evidencing significant differences between students with Maladaptive or Adaptive Academic profiles in these variables.

Additionally, the current study found a strict interplay between dispositional traits of neuroticism, anxiety, self-efficacy, and academic motivation consistent with the previous literature [20,21]. As a consequence, we retain that self-efficacy and academic motivation might act as protective factors in modulating the perceived stress for online learning activities under the uncertain condition of the COVID-19 pandemic. We found that students with a Maladaptive Academic Profile have significantly lower scores than students with an Adaptive Academic Profile in self-efficacy. Furthermore, self-efficacy helps individuals to develop the motivation to overcome stressful life situations [72] and shows a moderate-to-high correlation with some components of resilience, defined as the return to baseline function after exposure to traumatic life situations [73]. In this sense, self-efficacy may help the individual who faces a stressful life situation, such as the COVID-19 pandemic, to maintain the motivation to cope with it and not be damaged. From a theoretical point of view, self-efficacy helps students to maintain their academic motivation high, despite the difficulties they might face during their studies [74]. The pandemic has undoubtedly constrained students to adapt their learning strategies to a new stressful situation, and self-efficacy might have helped sustain their academic motivation [75]. Significant differences in the levels of academic motivation evidenced that Maladaptive Academic Profile students are more controlled than the Adaptive Academic Profile ones. To this respect, previous research on the relationship between motivation and mental health in university students highlighted how motivational aspects such as the importance of personal, academic, and life goals, commitment, and expectation in pursuit of them played a protective role against university students’ mental health-related risk-factors [76]. According to self-determination theory [77], autonomous regulated students who are committed to their life goals and feel scrutinized have a higher level of well-being since their needs for autonomy and competence are satisfied. Furthermore, the diathesis-stress theory suggests that students pursuing personally meaningful goals exhibit adaptive cognitive and behavioral strategies that help them overcome potential difficulties and reduce their risk of poor mental health [78]. In the academic context, students must face a delicate psychosocial transition to a professional career; hence, higher expectations of success and greater autonomous motivation predict higher well-being and lower stress [76]. On the contrary, university students with lower expectations of success and less optimism concerning social and achievement goals exhibited higher depression scores [79]. In the study by Piumatti [76], a positive motivation attitude protected against depressive feelings in university students. His findings suggest that the more students are highly positively motivated, the less they experience symptoms of psychological distress. The abovementioned aspects persisted and intensified during the COVID-19 pandemic. In line with [21], self-regulated behavior in university students might be considered a personal protective factor in academic stress management, as our results displayed.

The results of the current study should be interpreted regarding some strengths and limitations. A strength was identifying university students’ subgroups based on personality, trait and state anxiety, general self-efficacy, academic motivation, and fear of COVID-19. The literature on the university students’ condition during the COVID-19 pandemic is still heterogeneous across or within a series of socio-demographic, personal, or clinical variables. Our study contributes to bridging the gap derived by lacking homogeneous research because it allows splitting into clusters named Maladaptive Academic Profile and Adaptive Academic Profile, describing coherent profiles of students characterized by higher neuroticism, dispositional and situational anxiety, and fear of COVID-19 with lower self-efficacy and academic motivation in the first case or characterized by lower dispositional and situational anxiety and fear of COVID-19 with higher self-efficacy and regulated autonomous academic motivation in the second case. Furthermore, another strength can be found in studying the role of personality traits in influencing the capacity to cope with stressful events related to the academic profile of university students. Indeed, an under-explored question concerns the role of personality traits in rating the effects of personality traits in determining the psychological disposition toward learning in young adults. Nevertheless, future research is needed to remedy the limitations of this study. The most critical shortcoming lies in the methodology. First, a cross-sectional method was used, and no causal associations were tested nor investigated differences in the dependent variables before and after the COVID-19 pandemic. Second, we carried out an online questionnaire-based survey, and all variables were assessed through self-report measures. This choice was doubly motivated by the pandemic condition, the closure of face-to-face university courses, and the nature of the investigated constructs. In educational research, a commonplace is to measure autonomous motivation, effects, engagement, and self-efficacy with self-reports. Third, the study was conducted with a convenience sample that was non-representative of the Italian university student population. Demographic characteristics were not equally distributed, with more answers from girls. Notwithstanding the extensive dissemination of the online study through virtual classrooms and social media pages of students’ associations, the girls were probably more involved and responded more than boys. Future studies could consider longitudinally investigating changes over time and include larger samples and more objective measures to address these issues.

## 5. Conclusions

In sum, our study showed that the online learning activities during the COVID-19 pandemic represented a stressful situation for a consistent percentage of university students, having a specific psychological profile characterized by high neuroticism, trait/state anxiety, and fear of COVID-19 and low self-efficacy and autonomous regulated motivation. In this sense, we retain that the online learning activities during the COVID-19 pandemic have increased the vulnerability of students in their physical and mental health, which would have required professional support both for all the people who already suffered from health issues and for the general population [64,65]. It must be noted that the social restrictions imposed by the pandemic have limited people the possibility to ask for professional help. First, it might have happened that some psychologists did not recur to psychotherapy immediately after the pandemic emergency arose. Secondly, even though psychological support was available online, sharing a home with their own family could have discouraged people from engaging due to the fear of being listened to by other familiars [80]. Considering these barriers, it is not surprising that the most common supportive figures during the pandemic restriction were partners or familiars, as our findings show. Thus, the practical implications of the current study relied on the fact that the transition from offline to online education represented a risk condition for more vulnerable people. Psychologists, psychiatrists, and public stakeholders in education should consider implementing university counseling services to combat psychological distress related to the COVID-19 pandemic in the university population. Even if the emergency of the COVID-19 pandemic seems to have passed all over the world, its negative impact on university students’ physical and mental well-being could, in the long term, predispose them to the onset of psychiatric diseases [81]. Thus, as practical implications, it is hoped that universities offer counseling services to reinforce students’ protective factors, such as self-efficacy, and academic motivation, and at the same time monitor anxiety levels and personality traits which can lead to stressful and negative conditions.

## Figures and Tables

**Figure 1 ijerph-20-05255-f001:**
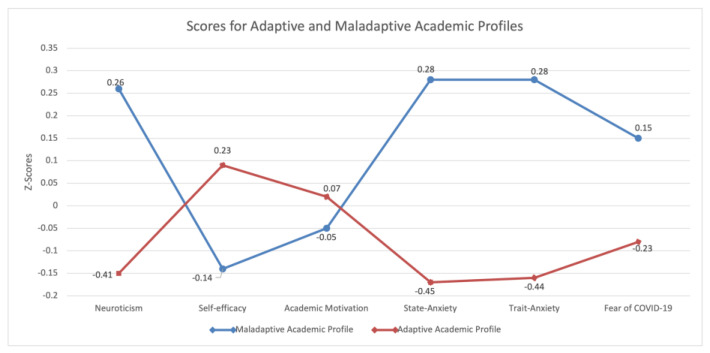
Z-scores for each studied variable for Adaptive and Maladaptive Academic Profiles.

**Figure 2 ijerph-20-05255-f002:**
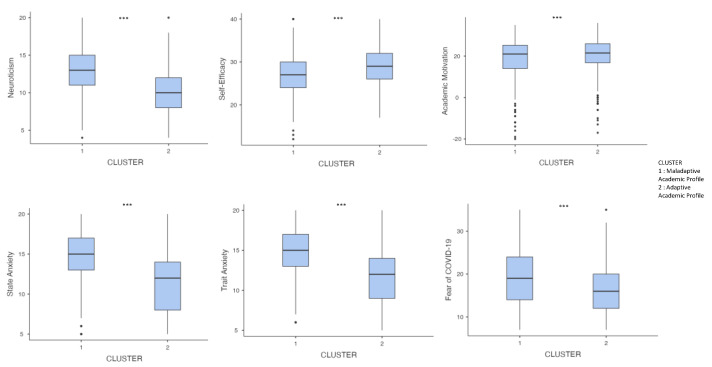
Plot of the between-group (Maladaptive and Adaptive Academic Profiles) differences on dependent variables. Note: *** *p* < 0.001.

**Table 1 ijerph-20-05255-t001:** Means scores, standard deviations, skewness, kurtosis, and Pearson’s bivariate correlations for study observed variables.

	1	2	3	4	5	6
1. Neuroticism	-					
2. Self-Efficacy	−0.489 **	-				
3. Academic Motivation	−0.255 **	0.262 **	-			
4. State Anxiety	0.513 **	−0.245 **	−0.186 **	-		
5. Trait Anxiety	0.640 **	−0.346 **	−0.191 **	0.672 **	-	
6. Fear of COVID-19	0.247 **	−0.110 **	0.422	0.257 **	0.269 **	-
Mean	11.23	28.02	19.51	12.66	12.87	17.46
SD	3.23	4.55	8.71	4.06	3.62	6.13
Skewness	0.07	−0.03	−1.24	−0.34	−0.24	0.34
kurtosis	−0.45	0.27	2.12	−0.69	−0.58	−0.49

Note: ** *p* < 0.010.

**Table 2 ijerph-20-05255-t002:** Auto-clustering.

Number of Clusters	Schwarz’s Bayesian Criterion (BIC)	Ratio of Distance Measures
1	16,534,885	
2	14,234,734	2142
3	13,475,259	1456
4	13,138,158	1446
5	13,086,876	1464
6	13,238,790	1498
7	13,536,109	1168
8	13,875,385	1056
9	14,227,911	1029
10	14,587,058	1185
11	14,982,234	1021
12	15,381,346	1004
13	15,781,153	1117
14	16,200,760	1014
15	16,622,636	1001

**Table 3 ijerph-20-05255-t003:** MANOVA results on study variables for Adaptive and Maladaptive Academic Profiles.

	Adaptive Academic Profile (*n* = 456)		Maladaptive Academic Profile (*n* = 572)		F	*p*	*η^2^p*
	**M**	**DS**	**M**	**DS**			
Neuroticism	12.59	3.02	10.15	2.98	166.7	<0.001	0.140
Self-Efficacy	26.96	4.71	28.87	4.24	46.45	<0.001	0.043
Academic Motivation	18.62	9.58	20.21	7.88	8.57	<0.01	0.008
State Anxiety	14.58	3.42	11.14	3.89	220.78	<0.001	0.177
Trait Anxiety	14.46	3.19	11.60	3.45	185.48	<0.001	0.153
Fear of COVID-19	19.00	6.41	16.23	5.62	54.02	<0.001	0.050

## Data Availability

Data are available under formal request to the Bioethics Committee of the University of Palermo.

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
