# Peer review of "The Switch to Online Learning during the COVID-19 Pandemic: The Interplay between Personality and Mental Health on University Students"

_ijerph, 2023, doi:10.3390/ijerph20075255_

Round 1

Reviewer 1 Report

The paper studied the impacts of Covid-19 pandemic on the mental health of university students switching from in-person class to online class using the two-step cluster analysis and other statistical analysis. The questionnaire was collected from March to May 2021 containing questions related to sociodemographic characteristics, personality, anxiety, self-efficacy, academic motivation, fear of Covid-19, and help request, etc. The design has some flaws: (1) The purpose was to examine the impact of Covid-19, but the questionnaire seemed not to ask students the changes of all factors before and after Covid-19 pandemic; (2) The results seemed intuitive even without Covid-19, students with Maladaptive academic profile are more likely to have higher neuroticism, dispositional and situational anxiety and fear, and lower self-efficacy and academic motivation, and those with Adaptive profile are opposite. It may be the cases, not really because of Covid-19, also because it had been one year when conducting the questionnaire, the Covid-19 impacts may be reduced.  (3) it seems the results had been reported in other studies, such as those elaborated in the second paragraph of Introduction. (4) The study said it used a well-suitable statistical technique to achieve stable and robust conclusions, but I did not see any evidence to show the conclusions are stable and robust. Also, the two-step cluster analysis seems to heavily rely on the distance, no definition of the distance, and the distance may change the number of clusters to be detected from the data.

Author Response

Response to Reviewer 1

(1) The purpose was to examine the impact of Covid-19, but the questionnaire seemed not to ask students the changes of all factors before and after Covid-19 pandemic;

Response: We appreciated this reviewer’s comment because it allowed us to specify better the aim of our paper.  Indeed, the main aim of the current study was to depict the psychological profiles of students forced to switch to online learning due to the COVID-19 pandemic, focusing also on the consequences on physical and mental health. We used a descriptive approach based on cluster analysis and not were interested in assessing the effects of different individual, situational, or learning variables before and after COVID-19. Accordingly, we better define our goal along the paper (lines 126-131; lines 136-140).

(2) The results seemed intuitive even without Covid-19, students with Maladaptive academic profile are more likely to have higher neuroticism, dispositional and situational anxiety and fear, and lower self-efficacy and academic motivation, and those with Adaptive profile are opposite. It may be the cases, not really because of Covid-19, also because it had been one year when conducting the questionnaire, the Covid-19 impacts may be reduced.

Response: To this point, we specify in the text: “It must be noted that the current study was performed in the second Italian lockdown phase (March-May 2021) which persisted in restricted measures of social distancing and isolation due to the COVID-19 pandemic. This choice allows us to evaluate the effects of more restrictive lockdown phase 1 on the studied dimensions, after one year, and identify how psychological variables such as personality, trait and state anxiety, self-efficacy, academic motivation, and fear of COVID-19 were related to the negative consequences of the COVID-19 pandemic on physical and mental health as to behavioral patterns to ask someone for help” (lines: 126-140).

(3) it seems the results had been reported in other studies, such as those elaborated in the second paragraph of Introduction.

Response: Even if previous studies reported similar findings, in the current study, we used clustering techniques to reach the purpose by identifying homogeneous subgroups presenting similar characteristics within a large sample considering that no study (to our knowledge) applied this technique (lines 136-140).

(4) The study said it used a well-suitable statistical technique to achieve stable and robust conclusions, but I did not see any evidence to show the conclusions are stable and robust. Also, the two-step cluster analysis seems to heavily rely on the distance, no definition of the distance, and the distance may change the number of clusters to be detected from the data.

Response: We reported on lines 300-304 the Auto-Clustering statistics method of SPSS we used to assess the optimal number of clusters in our analysis to give also more robustness to our analysis, and provided also a Table (lines 305). As well, we insert in the text the consideration that within the different methods used in the literature for determining the optimal number of clusters in a dataset, we choose the average silhouette method since it computes the average silhouette of observations for different values of k. The optimal number of clusters k is the one that maximizes the average silhouette over a range of possible values for k and for our data it seems that two clusters are the optimal number (lines 256-263).

Reviewer 2 Report

The overall aim of the present study was to separate subgroups of university students based on measures of personality, anxiety, general efficacy, academic motivation, fear of COVID-19, and nominal and ordinal measures of the impact of pandemic COVID-19 on physical and mental health, as well as requests for help, assessed in a large sample. In this study, a two-stage cluster analysis was applied, with an initial use of a distance measure to separate the groups, followed by a probabilistic approach to select the optimal subgroup model. The identified Cluster 1, , includes students who are neurotic and have high scores on dispositional and situational anxiety, as well as on the COVID-19 Anxiety scale. They have low levels of self-efficacy and are more extrinsically motivated. In cluster 2, there are emotionally stable students with low levels of dispositional and situational anxiety and COVID-19 anxiety. They have higher levels of self-efficacy and are more intrinsically motivated to achieve their academic goals.

This study confirms the association of psychological resilience with strong characteristics even in the face of a stressful event such as lockdown and remote working. 

The article presented is very interesting and presents a study with high statistical power 

Author Response

We appreciated the positive comments on our article and are very grateful to reviewer 2.

Reviewer 3 Report

This study is interesting and suits for the scope of the journal in that the study aims to analyze the impact of the COVID-19 pan-demic on the mental health of university students forced to switch their academic learning routines from face-to-face to online. However, as the authors mentioned in the manuscript already, all the findings were consistent with the results of the previous studies. In this regard, I strongly suggests the authors should present the differences and significance of the findings of this study compared to the findings of previous studies; otherwise this study remains at the surface level as the same study with other previous studies. Thus, it would be desirable for the authors emphasize the specific and unique findings of this study itself so that the findings of the study can be meaningful in the relevant field.

In addition, the implications of the study derived from the findings seem weak and vague. If the implications are strengthened by very specific and practical implications for the mental and physical health of university students.

Thank you.

Author Response

Response to Reviewer 3

  • I strongly suggests the authors should present the differences and significance of the findings of this study compared to the findings of previous studies; otherwise this study remains at the surface level as the same study with other previous studies. Thus, it would be desirable for the authors emphasize the specific and unique findings of this study itself so that the findings of the study can be meaningful in the relevant field. The implications of the study derived from the findings seem weak and vague. If the implications are strengthened by very specific and practical implications for the mental and physical health of university students.

Response: We are grateful to the reviewer for these suggestions since they allow us to improve the novelty of our study. To these points, we rewrite partially the Discussion focusing more on the interplay of multiple individual, social, and learning factors concurring together in the process of switching to online learning by university students.

  • Practical implication should reinforce students’ protective factors, as self-efficacy, working on academic motivation, and at the same time monitoring anxiety levels and personality traits which can lead to stressful and negative conditions.

Response: To this point, we define the practical implications of our study in the Conclusion (lines 520-523).

Round 2

Reviewer 3 Report

The authors revised and updated their manuscript accordingly based on the comments that I suggested.